# Single-Nuclei Transcriptome Profiling Reveals Intra-Tumoral Heterogeneity and Characterizes Tumor Microenvironment Architecture in a Murine Melanoma Model

**DOI:** 10.3390/ijms252011228

**Published:** 2024-10-18

**Authors:** Sushant Parab, Valery Sarlo, Sonia Capellero, Luca Palmiotto, Alice Bartolini, Daniela Cantarella, Marcello Turi, Annamaria Gullà, Elena Grassi, Chiara Lazzari, Marco Rubatto, Vanesa Gregorc, Fabrizio Carnevale-Schianca, Martina Olivero, Federico Bussolino, Valentina Comunanza

**Affiliations:** 1Department of Oncology, University of Torino, 10060 Candiolo, Italy; sushant.parab@unito.it (S.P.); federico.bussolino@unito.it (F.B.); 2Candiolo Cancer Institute, FPO—IRCCS, 10060 Candiolo, Italy; 3Department of Veterinary Science, University of Torino, 10095 Grugliasco, Italy

**Keywords:** melanoma, single-nuclei sequencing, transcriptome, heterogeneity, tumor microenvironment

## Abstract

Malignant melanoma is an aggressive cancer, with a high risk of metastasis and mortality rates, characterized by cancer cell heterogeneity and complex tumor microenvironment (TME). Single cell biology is an ideal and powerful tool to address these features at a molecular level. However, this approach requires enzymatic cell dissociation that can influence cellular coverage. By contrast, single nucleus RNA sequencing (snRNA-seq) has substantial advantages including compatibility with frozen samples and the elimination of a dissociation-induced, transcriptional stress response. To better profile and understand the functional diversity of different cellular components in melanoma progression, we performed snRNA-seq of 16,839 nuclei obtained from tumor samples along the growth of murine syngeneic melanoma model carrying a BRAFV600E mutation and collected 9 days or 23 days after subcutaneous cell injection. We defined 11 different subtypes of functional cell clusters among malignant cells and 5 different subsets of myeloid cells that display distinct global transcriptional program and different enrichment in early or advanced stage of tumor growth, confirming that this approach was useful to accurately identify intratumor heterogeneity and dynamics during tumor evolution. The current study offers a deep insight into the biology of melanoma highlighting TME reprogramming through tumor initiation and progression, underlying further discovery of new TME biomarkers which may be potentially druggable.

## 1. Introduction

The introduction of targeted therapies and immunotherapy has represented the most significant advances in the treatment of melanoma. However, the onset of resistance remains a challenge to overcome. Approximatively 50% of the patients have the activating missense mutation in the BRAF oncogene. BRAF and MEK inhibitors along with immune checkpoint inhibitors have improved patients’ outcomes [1,2,3,4]. These therapies extend patient survival but most patients will progress developing resistance. 

Melanoma heterogeneity makes treatment difficult due to the evolution of cell subsets that impact tumor growth, metastasis, and drug resistance [5]. Although the understanding of melanoma is rapidly improving with a better investigation of molecular heterogeneity, exploring new biomarkers might offer opportunities to develop prognostic models and advance therapeutic targets. A TME includes innate and adaptive immune cells, stromal cells, endothelial cells, cancer-associated-fibroblasts, nerves, and the extracellular matrix. Together, these components create a niche where tumor cells can grow and disseminate, playing a crucial role during tumor evolution. At the same time, a TME influences the onset of resistance to targeted therapy and immunotherapy [6]. However, the phenotypic identity and heterogeneity of tumor infiltrating stromal cells remains to be deeply characterized. 

Single cell sequencing is a powerful approach that has facilitated the analysis of cellular diversity and intercellular communication in tumors, as well as providing important biological insight into multiple cancer types, including metastatic melanoma, revealing cell states associated with resistance to targeted and/or immune therapies [7,8,9]. However, obtaining viable cells after a tissue dissociation procedure, without debris and environmental RNA molecules, is a prerequisite for single cell sequencing and represents a major limitation of this technology. Furthermore, enzymatic and/or mechanical tissue dissociation may alter the cell phenotype by inducing a transcriptional stress response, resulting in artifacts upon sequencing. Single-nuclei RNA-seq (snRNA-seq) is an emerging technology that circumvents the cell viability challenge by lysing cells to obtain the nucleus [10]. Basically, in contradiction to single-cell, single-nuclei sequencing can be performed on frozen archived samples [11]. 

In this study, we aimed to provide a comprehensive and global view of tumor heterogeneity and the TME landscape of melanoma using snRNA-seq as a profiling strategy that allows us to investigate the transcriptional states of heterogenous cell populations at high resolution. Based on the capabilities of the single-nuclei signaling, we tested the hypothesis that this approach could be used to identify both heterogeneity between melanoma cells and tumor-associated cell types, along with investigating the cell population dynamics during tumor progression. We therefore analyzed the single-nuclei transcriptional profiles of 16,839 nuclei from BRAF-mutated syngeneic melanoma tumors at two different stages of tumor growth at an early and an advanced stage.

This study offers important insights into melanoma biology, progression and identifies putative targets for therapeutic modulation. Investigation of the detailed composition of TME cells can provide evidence of additional biomarkers enhancing therapeutic approaches in melanoma. 

## 2. Results

### 2.1. Quality and Dimensionality of the Single-Nuclei and Sequencing Data

To explore the cellular heterogeneity and the tumor microenvironment in melanoma, single-nuclei transcriptome profiling was conducted on four tumor samples obtained along the progression of the D4M syngeneic melanoma model, carrying BRAFV600E mutation [12]. We used the tumors collected for a previous study [13], in which we described the in vivo growth of the D4M cell line by inoculating 500,000 cells. The tumors reached an average size of 200 mm^3^ and 2000 mm^3^ at day 9 and day 23, respectively, and were thus defined as early and advanced stage (Figure 1A and Appendix A). We performed high-quality snRNA-seq from four frozen tissue specimens, a biological replicate for each condition, that were collected more than 5 years ago and preserved in liquid nitrogen as indicated in the Methods section. The Nuclei were isolated with a Chromium isolation kit and loaded for snRNAseq (Figure 1B). After isolation, flow cytometry showed a pure preparation of a single nucleus with a negligible number of doublets (Figure 1C). Nuclei concentration, their integrity and quality were assessed by propidium iodine staining, microscope observation at 60× magnification and automated count by automated cell counter. Approximately 10,000 nuclei were partitioned in each snRNA-seq experiment using used 10X Genomics technology and sequenced (150 bp PE) by the Illumina NextSeq 500 platform. 

After classification of each barcode into cell and non-cell groups, there were an estimated 28,450 total nuclei sequenced. On average, we obtained more than 150G sequencing reads for each sample, with a median sequencing saturation of 91.40% (87.0–95.5%). We obtained approximately 3000 unique molecular identifiers (UMIs) for each nucleus, indicating sufficient coverage and transcript representations. The median number of genes detected per cell was 3042 in early stage and 3576 in advanced stages. To examine true nuclei, filtering parameters (feature threshold 200–300,000 and count threshold 50–50,000) were applied to eliminate empty droplets and doublets. After filtering for quality, we acquired a mean 3200 nuclei and 12,000 genes from every sample which were further used for downstream analysis, using a standardized pipeline (Figure 1D).

The snRNA-seq data from four samples (two specimens from early stage condition and two specimens from advanced stage condition) was analyzed by Seurat [14]. The integrated Seurat object consisted of 16,839 high-quality nuclei (7753 from early stage and 9086 from advanced stage of melanoma tumors (Figure 2A). Unsupervised Seurat-based clustering (see Methods) identified 21 clusters (c0–c20) which were further annotated into six major cell types. Initial cell type labeling on the population was performed by Seurat, followed by manual annotation. These six major cell types included melanoma cancer cells (n = 11987; 71%), myeloid cells (n = 3583; 21.2%), fibroblasts (n = 632; 4.0%), endothelial cells (n = 327; 2.0%), T lymphocytes (n = 272; 1.7%), and lymphatic endothelial cells (n = 38; 0.2%). A uniform manifold approximation and projection (UMAP) of these cells along with a feature dot plot highlighting a few marker genes from these clusters confirms our prediction (Figure 2A,C). To characterize different cell compositions at early and advanced stages, the proportions of each cluster associated with the main cell population was investigated. We found an overall increase in malignant cells and in the myeloid cell in the advanced stage while fibroblasts; T cells and endothelial and lymphatic endothelial cell populations decreased (Figure 2B and Appendix A). Flow cytometry data confirmed the significant reduction of CD4^+^ and CD8^+^ T cell population in advanced stage tumors compared to early stage D4M melanoma tumors (Appendix A). The cell-type assignment for each cluster was visualized as feature plots that assessed for the expression of cell type-specific canonical gene markers (Figure 2C), including the following: well-known melanocyte and melanoma marker *Pax3* and *Etv1* [15] for melanoma cancer cells; Adgre1, the gene coding for F4/80 and *Itgam* (Cd11b), for myeloid/macrophage cell populations; *Cdh11* and Loxl1 for fibroblasts; Cd247 (T-cell receptor T3 zeta chain), and *Il2rb* for T cells; *Cdh5* (VE-cadherin) and *Vwf* (Von Willebrand factor) for endothelial cells; *Flt4* for lymphatic endothelial cells (Figure 2C). In addition to these well-known markers, we also analyzed cluster-specific genes by differential gene expression analysis. The top 10 differentially expressed genes defining each cluster were plotted on a heatmap (Figure 2D). We found that melanoma cancer cells were characterized by the up-regulated expression of *Chic1*, *Cpe*, *Cdh6*, and *Tenm4*, while fibroblasts showed a high expression of *Chl1*, Grem2, Meg3, and *Rian*. Among the most differentially expressed genes in immune cell populations, we found *F13a1*, *Mrc1*, and *Msr1* for the myeloid cells and *Gzmf*, *Themis*, *Txk*, and *Itk* for the T Lymphocytes. On the other hand, endothelial cells were characterized by high expression levels of *Tacr1*, *Cyyr1*, *Shroom*, and *Fgd5*, and lymphatic endothelial cells were associated with the up-regulated expression of *Pcsk6*, *Ptx3*, *Cmah*, and *Plce1*. 

At the most granular level, we identified a total of 21 different cell clusters that are associated with different cell populations and major states. Clusters were numbered from 0 to 20 and contained a decreasing number of cells (c0-20). We found significant heterogeneity within melanoma cancer cells (11 clusters) and myeloid cell (5 clusters) (Figure 3A). No clusters specific to a single condition were observed (Figure 3B). 

To further explore the distinct cell composition in the melanoma cancer cells and TME across progression states, more detailed proportions were assessed. Particularly in advanced stage tumors, we observed enrichment in clusters c0, c1, c3, c7, c8, c9, c11, c15, c16, and c19, and inhibition in clusters c2, c5, c6, c10, c11, c12, c13, c14, c17, c18, and c20, while the proportion of clusters c14 and c19 were not affected by the tumor progression (Appendix A). 

### 2.2. Transcriptional and Functional Features of Melanoma Cancer Cells Reveal High Heterogeneity in D4M Tumors

Next, we focused on the transcriptional programs of subcategorized tumor cells. To distinguish the malignant cell populations, we scrutinized the transcriptional features between malignant cells in order to characterize the heterogeneity related to different melanoma clusters on a gene level. Firstly, we observed that clusters c0, c1, c3, c7, c11, c19, and clusters c2, c4, c5, c6, c14 are respectively up- and down-regulated along tumor progression (Figure 4A). Diversity within the clusters was identified by studying the cluster-specific genes (Figure 4B and Appendix A). We observed that cluster c0 had a high expression of collagen (*Col5a1* and *Col11a1*) with a range of genes involved in neuronal differentiation and synapse function and formation, like the postsynaptic adhesion molecule *Lrrc4c* (Netrin-G ligand-1), the glutamate receptor *Gria3*, and the axon guidance cue *Slit3*. Cluster c1 was instead characterized by metabolism and hypoxia-response. The top enriched genes in c1 included *Mt1* and *Mt2*, which encode metallothionein 1 and 2, the glycolytic enzyme *Pgk1* (phosphoglycerate kinase 1), and *Gpi1* (glucose 6-phosphate isomerase); and the cell death regulator, *Bnip3*. *Mt1* and *Mt2* play a role in detoxifications of heavy metals, are anti-oxidants and involved in tumor formation, progression and drug resistance [16]. *Bnip3* is highly up-regulated in hypoxia and contributes to the organization and dynamics of the cytoskeleton and focal adhesion complexes, regulating migration of melanoma cells and vasculogenic mimicry [17]. Cluster c2 was highly enriched for genes involved in DNA replication like *Pola1*, the gene that encodes for the catalytic subunit of DNA polymerase, *Diaph3* and *Prim2*, DNA primase, suggesting that this subset is associated with the proliferative state of D4M cells. Cluster c3 shows a high expression of different genes that promote the process of metastasization as *Spp1* (Osteopontin), *Serpinh1*, and *Mmp3*. Serpinh1 is a protein playing a role in collagen biosynthesis, that enhances the malignancy in osteosarcoma, but its role in melanoma is not well understood [18], while osteopontin and *Mmp3* are involved in cell adhesion, migration and in vivo tumor growth of melanoma [19,20]. Among the top up-regulated genes in cluster c4 we found collagens (*Col28a1* and *Col3a1*), *Fgfr1*, and transcription factor *Runx2*. Cluster c5 highly expressed *Dkk2*, a putative Wnt signaling inhibitor down-regulated in melanoma [21] and *Cdh13* (T-cadherin), an adhesion molecule that influences the migration and invasion of melanoma cells [22] and that functions as a pro-apoptotic tumor suppressor [23]. Cluster c6 was characterized by high expression levels of *Oxr1* (oxidation resistance 1) that prevents the formation of oxidative DNA damage and protects the genome integrity, and *Ank3* (Ankyrin-3) that instead acts as negative regulator of cell invasion [24]. Cluster c7 expressed high level of *Tead4*, a key mediator of the Hippo signaling pathway, implicated in the promotion of epithelial-mesenchymal transition and metastatic cascade, and the axon guidance cue, *Sema3a*, implicated in tumor angiogenesis [25]. Cluster c11 showed an increased expression of *Ptgs2* (cyclooxygenase-2) and the matricellular protein *Sparc*, a critical tumor-derived mediator of vascular permeability during melanoma metastatic dissemination [26]. We found also high level of *Lockd* and *Efcab11* in cluster c14 and *Ifi203*, *Stat1*, and *Parp14* in cluster c19. Using the markers identified for each melanoma cancer cell clusters, a gene set enrichment analysis was performed using a hallmark pathway to identify variations among clusters (Figure 4C). Interestingly, among the clusters associated with melanoma cancer cells, three clusters (c2, c6, c14) showed up-regulation of “G2M transition”, “mitotic spindle” and “E2F targets”, while the other seven clusters (c1, c2, c3, c5, c6, c7, c11) showed up-regulation of “MYC targets v1” and “mTORC1”. Moreover, we found that cluster c1 was the only cluster with down-regulation of “NOTCH signaling”. Clusters c1 and c11 showed up-regulation of glycolysis, and cluster c1 was also characterized by up-regulation of the “hypoxia response”. GSEA also revealed that the angiogenesis pathway was overrepresented in some melanoma clusters (c0, c2, c3, c4 and c11).

Collectively these results suggest that clusters c2, c6, and c14 consist of actively dividing tumor cells driving tumor expansion, while cluster c1 characterizes cells undergoing hypoxia and death. 

### 2.3. Gene Expression Heterogeneity in Myeloid Population in Melanoma Progression

To identify the TME landscape dynamics during melanoma progression, we evaluated the myeloid cell distribution and heterogeneity across early and advanced stage tumors. We recovered a total of 3583 myeloid cell transcriptomes, which included 1521 from early stage and 2062 from advanced stage. Altogether, the myeloid population represented the largest population in the TME, constituting 67% of non-malignant cells. The snRNA-seq analysis identified six distinct transcriptional states in myeloid population: c8, c9, c12, c13, c15, and c16 (Figure 3A). Cell fraction analysis suggested that clusters c12 and c13 were enriched in early-stage tumors (c12: 26.4% vs. 8.3%; c13: 18.5% vs. 7.7%), while c8, c9, c15, and c16 were more frequent in advanced stage tumors (c8: 29.9% vs. 22.9%; c9: 27.8% vs. 17%; c15: 14.2% vs. 7.4%; c16: 12.1% vs. 7.8%) (Figure 5A). We then investigated the expression pattern of typical myeloid markers: *Ptprc*, *Adgre1*, *Itgam1*, *Lyz2*, *Cd84* (pan-macrophage markers); *Cd38*, *Cd86*, *Cxcl10*, *Il1b*, *Tnf*, *Ciita*, *Cd74*, *Cd40*, *Stat1* (M1-profile); and *Arg1*, *Klf4*, *Mrc1*, *Socs2*, *Tgm2* (M2-profile) (Figure 5B). It should be noted that recovered myeloid cells subclusters were not clearly representative of the previously reported M1-like/M2-like phenotypes and that most of these genes displayed heterogenous or bimodal expression patterns within populations and an overlap between populations indicating a spectrum of intermediary phenotypes with putative pleiotropic functions. Therefore, from this gene expression, no clear patterns emerged that could allow assigning these populations to previously proposed M1 and M2 macrophage subsets. To shed light on the complexity of the myeloid cells in melanoma D4M tumors, we investigated cell type-specific transcriptional changes during tumor progression. We examined genes showing the most significant enrichment in expression in each cell population (Figure 5C and Appendix A). Macrophages of cluster c8 revealed a high expression of Ms4a4a, a member of the membrane-spanning four domain family, and Stab1 (Stabilin-1), a homeostatic scavenger receptor. Ms4a4a characterizes M2 macrophages and it has been suggested to have an immunotherapeutic potential [27]. Stabilin-1 defines a subset of macrophages with a role in the regulation of tissue fibrosis [28]. Myeloid cells of cluster c9 revealed a high expression of Pparg, peroxisome proliferator-activator receptor gamma and Ipcef1 (Interaction protein for cytohesin exchange factors 1). Pparg is involved in monocyte maturation to macrophages and in regulation of inflammation [29]. Furthermore, it suppresses the immunoreactive state of macrophages and promotes an immunotolerant state marker [30]. 

Myeloid cells of cluster c12 disclosed the high expression of the following: *Nav3* (Neuron navigator 3) [31] which is involved in cell migration; *Axl*, a tyrosine kinase receptor activated in macrophages in response to tissue injury and mediating the clearance of apoptotic cells [32] and the regulation of the cytokine; and *Osbpl6* (oxysterol binding protein-like 6), which is transcriptionally regulated in a subset of macrophages in response to cholesterol loading [33]. 

Macrophages of cluster c13 showed a high expression of the chemokine receptor Ccr2 and of the serotonin receptor *Htr7*. CCR2 expressing macrophages have phenotypical similarities to M1-polarized macrophages, with pro-inflammatory phenotypes. Htr7 mediated the inhibitory action of serotonin on the release of pro-inflammatory cytokines in macrophages and stimulates macrophage polarization towards the acquisition of an profibrotic phenotype [34]. Myeloid cells of cluster c15 revealed a high expression for *Slca1* gene coding for GLUT1 glucose transporter among other genes involved in glycolytic pathway (*Hk3*, *Pfkfb3*, *Hk2*, *Pfkp*, *Eno1*; Appendix A) and *Vegfa*. High expression of the proliferation marker and the DNA glycosylase Neil3 characterized cluster c16, suggesting that this myeloid subset might be characterized by growing cells. 

To further apprehend the potential functional heterogeneity of these five myeloid subsets, we performed gene ontology (GO) term analysis. As expected for cells sharing a similar myeloid lineage, some biological processes were clearly overlapping, including processes related to inflammatory responses, innate immunity, and defense response. Using all genes in the mouse genome database as a reference, a statistical overrepresentation test was performed on the significantly up-regulated genes from each myeloid cluster (clusters c8, c9, c12, c13, c15, c16) to identify enriched pathways based on the GO biological process (Appendix A). Using a false discovery rate (FDR) cut-off of 0.05, the top twenty pathways for each cluster are shown in Figure 5D. An enriched GO biological process in cluster c8 included several pathways involved in myeloid cell differentiation to complement activation and regulation of tumor necrosis factor production. Cluster c9 was enriched for positive regulation of interleukin-1 b production and phagocytosis. Up-regulated pathways in cluster c15 were instead related to the negative regulation of immune system processes, like the negative regulation of mast cell activation, T cell cytokine production, and leukocyte cell-cell adhesion. At the same time, cluster 15 was enriched for the arginine catabolic process and the regulation of T cell anergy. The up-regulated GO biological processes in cluster 16 included synapses pruning and cell junction disassembly, phagocytosis, and B cell-mediated immunity. Interestingly, the two macrophage subpopulations, c12 and c13, that were inhibited during tumor progression showed enrichment in pathways with anti-tumoral and immune supportive properties. Cluster c12 up-regulated GO biological processes, including lymphocyte activation and differentiation, cytokine-mediated signaling pathway, dendritic activation, and response to IL-6. Cluster c13 was enriched for pathways related to regulation of antigen processing and presentation through MHCII (major histocompatibility complex class II) and positive regulation and activation of T cells. 

Finally, we investigated cell–cell interaction based on ligand–receptor expression levels using a CellChat bioinformatic tool [35]. CellChat identified 292 significant (FDR < 0.05) ligand–receptor interactions among the 21 different cell clusters (Figure 6A), which were further categorized into 75 signaling pathways (Appendix A). It also highlighted the heterogeneity within the myeloid cell subpopulations, where cluster c12 was found to show stronger, and a maximum number of, interactions with all the remaining 21 clusters, as compared to the other myeloid cells (Figure 6B). Interestingly, cluster c12 was also associated with the myeloid cells subset that interacts the most with T cells (cluster c18) (Figure 6C). Specific and unique cell–cell communication interaction between the c12 myeloid cells and T cells identify thrombospondin, netrin, sema7, collagen, and laminin pathways (Figure 6D and Appendix A).

## 3. Discussion

Melanoma is one of the most heterogeneous human cancers that exhibits a high level of biological complexity during disease progression. Failure of the treatment or acquired resistance are in part attributable to intratumoral heterogeneity not only of cancer cells but of TME populations, particularly in response to BRAF and anti-PD1 therapy [36,37,38,39]. Therefore, it is important to better elucidate the subpopulations of the immune system, endothelial and stromal cells among the malignant cells that compose tumors. 

The development in single-cell omics technology opened up the possibility to map cellular heterogeneity and recover cellular identities independent of prior defined labelling strategies [40], uncovering previously unrecognized cell populations or functional states, markers, and potential molecular regulators [41,42,43]. To date, most studies have been conducted on fresh tissue specimens, the use of which poses significant practical challenges. The exploitation of snRNA-seq may overcome these limits and to our knowledge this is the first study to utilize single-nuclei RNA-seq to characterize the molecular landscape of a treatment-naïve murine melanoma model. Gene expression sequencing of single-nuclei differs from single-cell in the information it provides. Single-nuclei sequencing captures polyadenylated RNA transcripts that are actively being transcribed in the nucleus, whereas single-cell sequencing captures all polyadenylated RNA within the cell’s cytoplasm. Despite these differences, single-nuclei sequencing has been shown to provide equivalent gene detection signatures and accurate cell identification while minimizing bias compared to single-cell sequencing [9,10,44,45]. It has been previously reported that two main malignant phenotypes coexist in melanoma that are distinguished on the base of gene expression: a proliferative state, that is weakly motile, and a metastatic profile that shows a transcriptional profile similar to neural-crest transcriptome. Transcriptional reprogramming of proliferative melanoma cells into a phenotypically distinct invasive cell subpopulation is a critical event at the origin of metastatic spreading [8,46,47]. We identified 11 different major malignant cell subpopulations, among which their proportions dynamically change during tumor progression. According to this, the clusters enriched during tumor progression of D4M melanoma were associated with a neural-crest stem cell-like state. On the other hand, clusters associated with cell cycle and proliferation were inhibited in advanced stage tumors. These could suggest a specific transcriptional transition towards a state characterized by increased cell motility and aggressiveness. Collectively, the enriched hallmark results support the clustering and annotation results by confirming shared relationships and functions among common cell types, and aid in the elucidation of the melanoma cancer cells heterogeneity. 

Our data also provides insight into the composition of TME. Generally, innate immune cells are the most abundant stromal components in melanoma TME [48]. We reported that myeloid cells are the most abundant TME population in D4M melanoma. Macrophages are plastic cells often acquiring opposite functions according to external cues and metabolic changes. Macrophages are broadly divided into M1 and M2 subtypes. M1 macrophages express high levels of pro-inflammatory cytokines and promote a pro-inflammatory milieu; whereas M2 macrophages release anti-inflammatory molecules, suppress antitumor immunity, stimulate angiogenesis, and promote tumor invasion. However, this dichotomous classification is too simple, as the in vivo environment of macrophages is more complex [49]. Therefore, a more precise evaluation is needed for accurate characterization of the dynamics of macrophage heterogeneity in a TME. Using snRNA-seq sequencing to investigate the landscape in melanoma tumors we uncovered six major myeloid populations and established their gene expression signature. The six identified macrophage populations displayed a common expression of both M1 and M2 gene markers and were not clearly representative of the previously reported polarization state phenotypes. We found that one of the subsets, inhibited in the advanced stage tumor cluster c15, was associated with the function of antigen presentation. One of the mechanisms associated with the resistance to immunotherapy is related to deficiency in antigen presentation [50]. Accordingly, we previously observed that the D4M melanoma model is not sensitive to the anti-PD1 antibody. At the same time we found that a shift towards a more favorable TME landscape, through vessel normalization, activation of M1 macrophages, and CD8+ T cells recruitment was elicited by vascular endothelial growth factor A (VEGFA) removal and BRAF inhibition, and boosted the immune checkpoint blockade [13]. VEGFA is a key promoter of both tumor angiogenesis and immunosuppression, and is an attractive target for combinatorial cancer therapy [51]. Interestingly, we found that the specific subset of myeloid cells c15 enriched during tumor progression, expressed high level of VEGFA, suggesting that the resistance mechanism to immunotherapy might be led by this specific subpopulation of myeloid cells and reverted by the neutralization on VEGFA with specific antibodies. The results here reported suggest a determinant role for the genes specifically modulated in different macrophage subsets along tumor progression. Therefore, it is intriguing that there is need for a combination of an anticancer therapy with corresponding modulators of macrophage phenotype. Further studies should focus on validating these findings across defined patient subgroups. In future, tailoring TAM-targeted therapies in combination with other therapeutic strategies may constitute a promising alternative treatment for patients. 

### Limitations of Our Study

Despite our promising observations suggesting that snRNA-seq technique might improve our understanding of melanoma biology, more detailed insights are required to better profile the progression of tumors. Our snRNA-seq analysis performed on four samples simply demonstrated a clear change in TME cell populations in early and advanced stages of syngeneic D4M in vivo growth. It is plausible that more samples could detect subtler differentiation within TMEs including the detection of minute cell populations. Furthermore, any clinical translation of the observations here reported will require validation in different progression stages of human melanomas. 

## 4. Materials and Methods

### 4.1. Sample Description

Melanoma tumor specimens were obtained from a D4M syngeneic melanoma model, as described in [13]. All animal procedures were approved by the Italian Ministry of Health (protocol 21635.13) and were performed in accordance with institutional guidelines and international law and policies. Freezing of tumor samples was performed as quickly as possible after sample collection using a standard biobanking technique. Tumor specimens were diced into approximately 50 mg pieces, washed in PBS and then placed in cryo-tubes containing a cryopreservation medium (90% FBS, 10% DMSO). The tubes were then transferred to −80 °C for 48h and in liquid nitrogen for long-term storage. Samples used in this work were stored in liquid nitrogen for approximately 5 years prior to nuclei isolation. 

### 4.2. Nuclei Isolation

Nuclei were isolated from frozen tissue by following the Chromium Nuclei Isolation Kit protocol (CG000505, 10X Genomics, Pleasanton, CA, USA). The nuclei isolation protocol was followed according to the manufacturer’s directions. Nuclei were visualized and counted using propidium iodide and the automated cell counter Countess II FL (Thermo Fisher Scientific, Waltham, MA, USA) to determine the concentration for the library preparation. 

### 4.3. Droplet-Based Single-Nuclei Sequencing

A total of 10,000 nuclei were loaded per single-nuclei RNA sequencing sample. Single-nuclei barcoding and library construction were performed using Chromium Next GEM Single Cell 3′ Kit v3.1 on 10X Chromium Controller (10X Genomics, Pleasanton, CA, USA) using the manufacture-recommended default protocol and settings. Due to the number of loaded nuclei, the cDNA amplification was obtained with 11 cycles PCR. and cDNA quality was assessed using the 2100 Bioanalyzer (Agilent). Samples were sequenced on NextSeq500 sequencer (Illumina Inc., San Diego, CA, USA) generating 28bp read1 and 90bp read2 data. Resulting FASTQ files were processed using CellRanger analysis suite for alignment to the mm10 (Mus musculus) reference genome, identification of empty droplets and determination of a count threshold. All downstream analyses were performed in Seurat using a default pipeline.

### 4.4. Processing of the Single-Nuclei Sequencing Data

The raw base call (BCL) files generated by sequencer were converted to FASTQ format by using Cell Ranger (v7.1.0) command “mkfastq”. These demultiplexed FASTQ files were further aligned by Cell Ranger “count” command (with introns included) to the mouse mm10-2020A genome. Loupe Browser was used to perform an initial quality check on all the samples, and for identification of the clusters and gene expression matrices.

### 4.5. Filtering the Background Noise and Quality Control

We used the “remove-background” function from CellBender (v0.3.0) to remove the technical artifacts and empty droplets from the gene expression data. The ‘raw_feature_bc_matrix.h5′ file from Cell Ranger was used as an input for CellBender. The filtering was performed individually on every sample, before going ahead with the quantification.

### 4.6. Quantification and Identification of the Cell Types

The processed expression matrix from CellBender was directly uploaded in Seurat (v5.0.1.9004) to check for the quality of the nuclei. Filters were applied to keep the nuclei with 500–4000 genes, <20% of mitochondrial reads and <4% of ribosomal reads in every sample. Filtered matrices were then normalized using the ‘LogNormalize’ method, and the top 2000 variable genes were identified by the ‘FindVariableFeatures’ (using the ‘vst’ method). The matrices were further centered by the ‘ScaleData’ function and projected for principal component analysis (PCA) as well as uniform manifold approximation and projection (UMAP). Preliminary cell type annotations were performed using SingleR (v2.6.0) against the built-in reference ‘MouseRNAseqData’.

### 4.7. Integration of Individual Samples and Differential Gene Expression

All samples were integrated into Seurat. The ‘FindIntegrationAnchors’ function selected 2000 anchors between the samples using the top 30 dimensions. These pre-computed anchors were used to integrate the data by the ‘IntegrateData’ function. PCA and UMAP dimension reduction based on the top 30 principal components were performed. ‘FindNeighbors’ and ‘FindClusters’ were performed on the dataset before doing any manual annotation. The ‘FindAllMarkers’ function from Seurat identified differentially expressed genes and overexpressed (positive) markers in each population.

### 4.8. Enriched Pathways Analysis 

Gene set enrichment analysis using hallmark and canonical pathways in *mus muscuslus* genome was accomplished using the R package escape (v.2.0.0). To identify enriched GO biological processes among clusters, PANTHER (v19.0) was used to perform a statistical overrepresentation test (Fisher’s exact test with FDR correction) using the GO Ontology database

### 4.9. Flow Cytometry 

T cells tumor infiltration was performed with staining at 4 °C for 20 min with the following antibodies: anti-mouse CD45 (clone 30-F11; Biolegend, San Diego, CA, USA), anti-mouse CD4 (clone GK1.5; BD Biosciences, Milano, Italy), and anti-mouse CD8 (clone 53-6.7; BD Pharmigen). Cells were detected using the Cyan ADP flow cytometer (Beckman Coulter) and data were analyzed with the Summit 4.3 software (Beckman Coulter). Quadrants were set based on isotype control antibody and cells were gated among total DAPI^-^ cells. 

### 4.10. Statistical Analysis

Statistical analyses of snRNA-seq data were performed by CellBender, Seurat (v5.1.0) and CellChat. Most of the analyses were performed in Python (version 3.11.7) and R (version 4.4.0) software. And *p*-value below 0.05 was considered statistically significant in this research. Flow cytometry analysis was performed by Student’s *t*-test. 

## Figures and Tables

**Figure 1 ijms-25-11228-f001:**
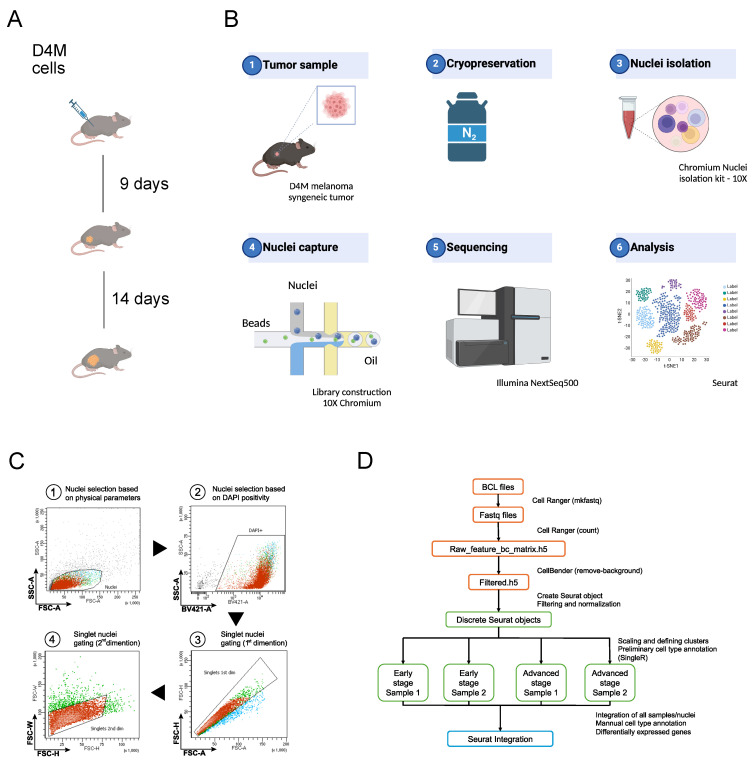
Schematic diagram of study design workflow (**A**) Syngeneic D4M melanoma model. (**B**) Graphical representation of the experimental setup for nuclei purification and sequencing. The figure was created with BioRender. (**C**) Gating strategy for FACS isolation of a single nucleus isolated using Chromium Nuclei Isolation Kit. Single DAPI⁺ events were considered nuclei. (**D**) Computational pipeline.

**Figure 2 ijms-25-11228-f002:**
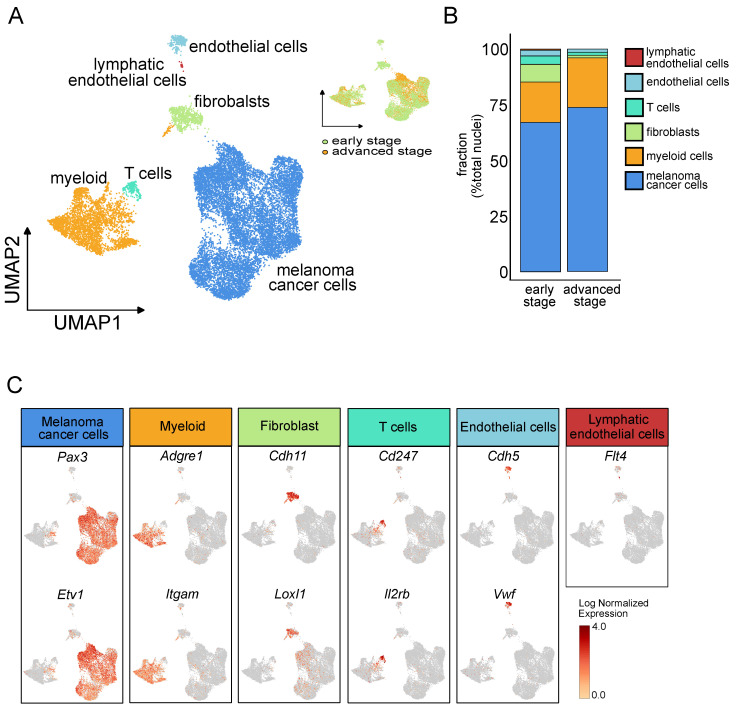
snRNA-seq of melanoma syngeneic tumors identifies main cell populations: (**A**) Left panel, clustering of 16,839 high-quality nuclei from D4M melanoma syngeneic tumor samples (n = 4) represented on a two-dimensional Uniform Manifold Approximation and Projection (UMAP) plot and grouped into six major cell types. Upper right panel, UMAP plot showing the distribution originating from early stage (n = 2) or advanced stage tumor samples (n = 2). (**B**) Bar plot showing the fraction of each cell type (melanoma cancer cells, myeloid cells, fibroblasts, T cells, endothelial cells, lymphatic endothelial cells) according to the origin samples, early stage or advanced stage tumors. (**C**) Feature plot assessing the gene expression levels of the selected cell-type specific marker genes: *Pax3* and Etv1 (melanoma cancer cells); Adgre1, Itgam (myeloid/macrophage cell population); Cdh11 and Loxl1 (fibroblasts); Cd247 and Ilr2b (T cells); Cdh5 and Vwf (endothelial cells); Flt4 (lymphatic endothelial cells). Gene expression patterns are projected onto UMAP. Scale: log-transformed gene expression. (**D**) Heatmap showing the top five differentially expressed genes in each cluster indicating the main cell populations. Clusters are identified on the left y-axis and gene symbols are listed on the top x-axis. Red indicates up-regulation and blue indicates down-regulation. Scale: log2 fold change.

**Figure 3 ijms-25-11228-f003:**
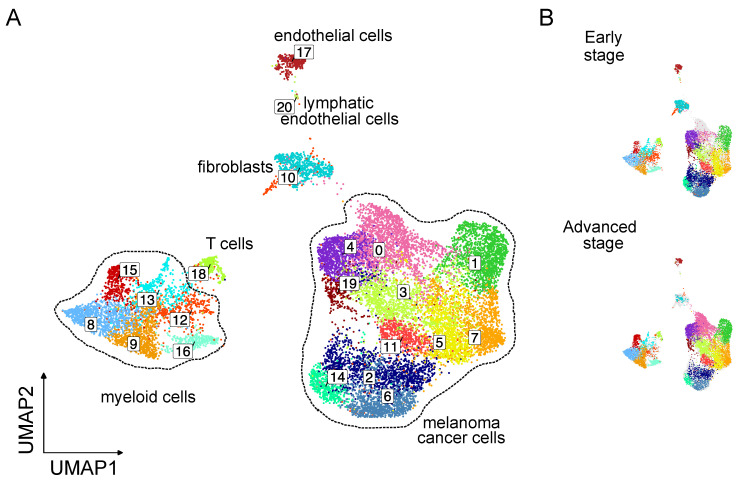
snRNA-seq of murine melanoma D4M tumors identifies 21 different cell clusters. (**A**) snRNA-seq of nuclei isolated from murine melanoma D4M tumors (n = 4). Dimensionality reduction and identification of clusters of transcriptionally similar cells were performed in an unsupervised manner using Seurat package. (**B**) UMAP plot showing the distribution originated from early stage (n = 2) or advanced stage tumor samples (n = 2).

**Figure 4 ijms-25-11228-f004:**
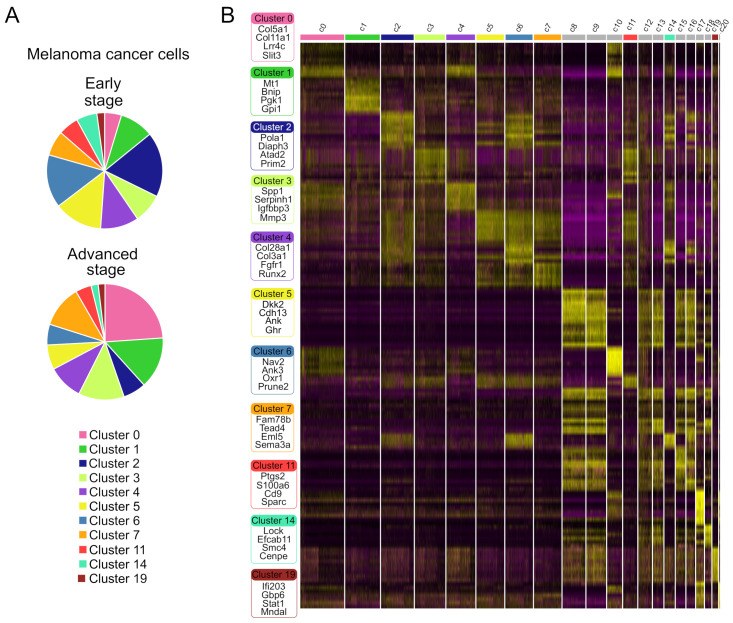
snRNA-seq reveal heterogeneity in melanoma cancer cells in murine melanoma D4M tumors. (**A**) Bar plot showing the fraction of each cluster (c0, c1, c2, c3, c4, c5, c6, c7, c11, c14, c19) associated to melanoma cancer cells within the total melanoma cancer cells, according to the origin samples, early stage (upper panel) or advanced stage tumors (bottom panel). (**B**) Heatmap showing the top 20 most up-regulated genes (ordered by decreasing *p* value) in each cluster defined in Figure 3A and selected enriched genes used for biological identification of melanoma cancer cells heterogeneity among each cluster associated to melanoma cancer cells. Scale: log2 fold change. The top bars in color label corresponding to melanoma cancer cell clusters. Top bars in grey indicate clusters associated with a TME. Normalized gene expressions are shown. Full gene list for each cluster can be found in Appendix A. (**C**) Gene set enrichment analysis among melanoma tumor cell clusters using hallmark pathways. Heatmap of the enrichment scores produced from gene set enrichment analysis using hallmark pathways. Red indicates up-regulation and blue indicates down-regulation. Clusters are indicated on the x-axis.

**Figure 5 ijms-25-11228-f005:**
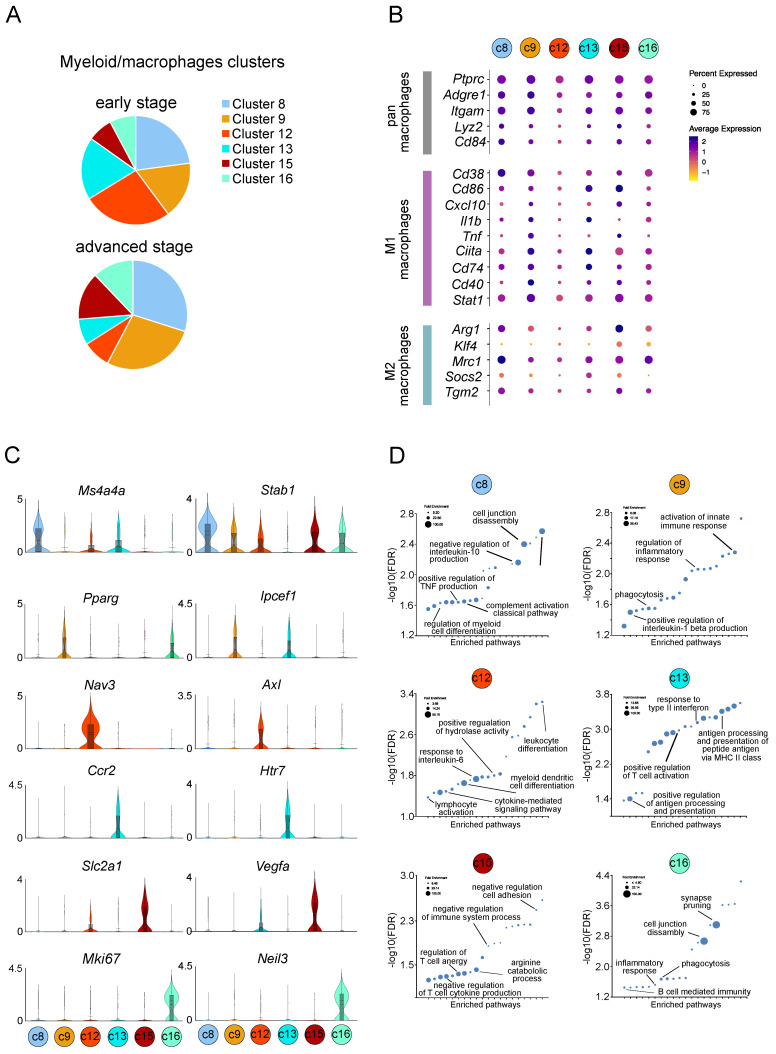
snRNA-seq reveals heterogeneity in myeloid/macrophage cells in murine melanoma D4M tumors. Gene expression signature of macrophages in murine D4M melanoma model. (**A**) Bar plot showing the proportion/fraction of each cluster (c8, c9, c12, c13, c15, c16) associated to myeloid cells within the total myeloid cells according to the origin samples, early stage (upper panel) or advanced stage tumors (bottom panel). (**B**) Dot plot heatmap showing the gene expression level of pan-macrophage, M1 macrophage, and M2 macrophage markers among clusters associated with myeloid cells (c8, c9, c12, c13, c15, c16). The color intensity of each dot represents the average level of marker gene expression, while the dot size reflects the percentage of the cells expressing the marker within the clusters. (**C**) Violin plots of log-transformed gene expression of selected genes showing statistically significant up-regulation in the indicated clusters associated with myeloid cells. (**D**) Gene ontology of differentially expressed genes among clusters associated with myeloid cells: c8, c9, c12, c13, c15, c16. The top 20 enriched GO biological processes and their associated fold enrichment and false discovery rate (FDR) are shown. Dot size correlates to the corresponding fold enrichment. A full report can be found in Appendix A.

**Figure 6 ijms-25-11228-f006:**
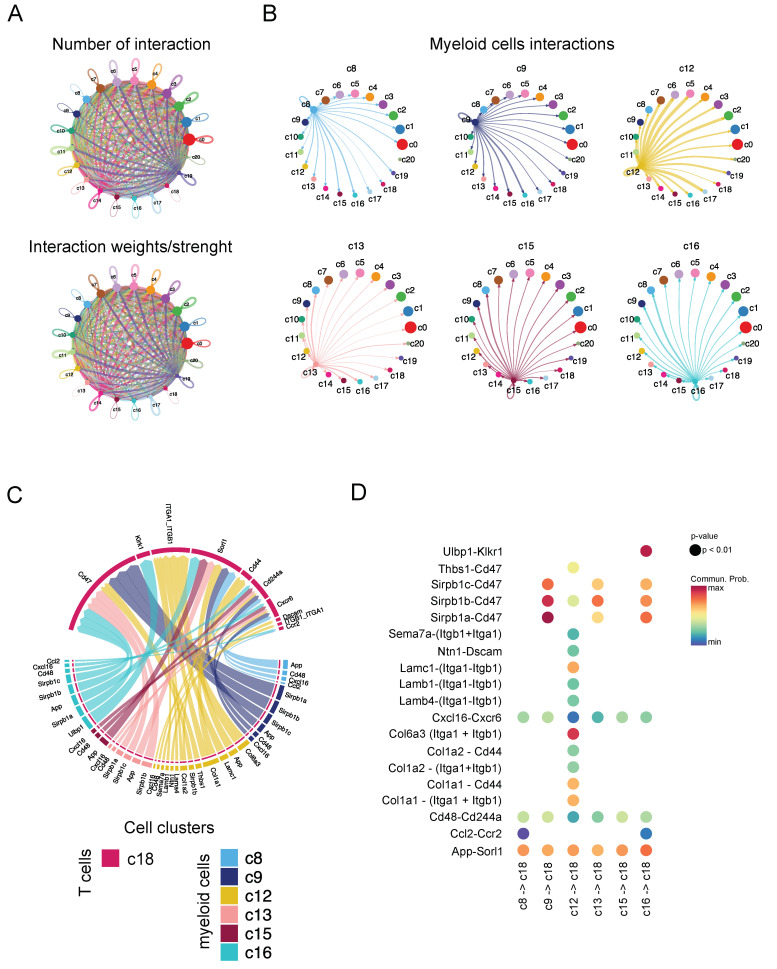
CellChat analysis reveals cell–cell interactions in murine melanoma D4M tumors. (**A**) Total number of interactions identified in every cluster predicted by Seurat integration. Strength or weight of every interaction which further confirms few clusters that have the maximum strong interactions within different cell populations. (**B**) Interactions found in different/individual myeloid clusters, from which cluster c12 has the maximum stronger interactions as compared to others. (**C**) Cell–cell interactions between different myeloid clusters and T cells. (**D**) Bubble plot highlights some unique ligand–receptor interactions between a myeloid cluster (c12) and T cells (c18).

## Data Availability

All raw sequencing data generated in this study have been submitted to the NCBI Gene Expression Omnibus (GEO; https://www.ncbi.nlm.nih.gov/sra/PRJNA1159010) accessed on 1 November 2024 and are available under the BioSample accession numbers SAMN43611309, SAMN43611310, SAMN43611311, SAMN43611312. Until 01 November 2024, requests to access the datasets should be directed to Sushant Parab, sushant.parab@unito.it.

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
