# Peer review of "Single-Nuclei Transcriptome Profiling Reveals Intra-Tumoral Heterogeneity and Characterizes Tumor Microenvironment Architecture in a Murine Melanoma Model"

_ijms, 2024, doi:10.3390/ijms252011228_

Round 1

Reviewer 1 Report

Comments and Suggestions for Authors

In this study, the authors employed single nucleus RNA sequencing (snRNA-seq) technology to investigate the tumor microenvironment in murine melanoma model. This study is well designed, experiments were adequately performed and results are solid. Although the data presented in this study is largely descriptive and further experimental validations are warranted, this study clearly demonstrated the advantage of snRNA-seq in analyzing tumor heterogeneity. I suggest several minor revisions which need to be addressed before acceptance.

In Figure 2, there are two subpanel “B”. The latter one may be corrected to “C”.

In Figure 3A, cluster numbers are too small to read.

Page 6, line 166: “To further explore the distinct cell composition in the melanoma cancer cells and TME 166 across progression states, more detailed proportions were assessed (Table S2).” Please add brief description what findings were obtained from the data in Table S2.

In Figure 5D, labels in X-axis are missing.

Page 12, line 313: “Table 5” may be corrected to “Table S5”.

Pages 13-14: Sentences in lines 384 to 394 and those in lines 396 to 406 are completely same. It may be an error during editing the manuscript, and should be corrected.

Author Response

Response to Reviewer #1

  1. Summary

In this study, the authors employed single nucleus RNA sequencing (snRNA-seq) technology to investigate the tumor microenvironment in murine melanoma model. This study is well designed, experiments were adequately performed and results are solid. Although the data presented in this study is largely descriptive and further experimental validations are warranted, this study clearly demonstrated the advantage of snRNA-seq in analyzing tumor heterogeneity. I suggest several minor revisions which need to be addressed before acceptance.

Thank you for your positive assessment of our manuscript entitled “Single-nuclei transcriptome profiling reveals intra-tumoral heterogeneity and characterizes tumor microenvironment architecture in a murine melanoma model” (ID: ijms-3246868).  We have considered all the suggestions carefully and have made correction which we hope meet with approval. Revised portion are marked in yellow in the paper. Main corrections and the answers to your comments are shown below.

  1. Questions for general evaluation

Does the introduction provide sufficient background and include all relevant references?

Yes/Can be improved/Must be improved/Not applicable

Are all the cited references relevant to the research?

Yes/Can be improved/Must be improved/Not applicable

Is the research design appropriate?

Yes/Can be improved/Must be improved/Not applicable

Are the methods adequately described?

Yes/Can be improved/Must be improved/Not applicable

Are the results clearly presented?

Yes/Can be improved/Must be improved/Not applicable

Are the conclusions supported by the results?

Yes/Can be improved/Must be improved/Not applicable

  1. Point-by-point reponses to Comments and Suggestions for Authors

Point 1. In Figure 2, there are two subpanel “B”. The latter one may be corrected to “C”.

Response 1. Thank you for the advice, we corrected the third panel to “C”

Point 2. In Figure 3A, cluster numbers are too small to read.

Response 2. Thank you for the suggestion, we increased the font size that identify the cluster numbers.  

Point 3. Page 6, line 166: “To further explore the distinct cell composition in the melanoma cancer cells and TME 166 across progression states, more detailed proportions were assessed (Table S2).” Please add brief description what findings were obtained from the data in Table S2.

Response 3.

Thank you very much for your advice. We added the sentence (Now, Page 6, line 170-173) “In particular, in advanced stage tumors we observed the enrichment in clusters c0, c1, c3, c7, c8, c9, c11, c15, c16, c19 and the inhibition in clusters c2, c5, c6, c10, c11, c12, c13, c14, c17, c18, c20, while the proportion of clusters c14 and c19 were not affected by the tumor progression”.

Point 4. In Figure 5D, labels in X-axis are missing

Response 4. Thank you for the kind advice, we added the X-axis lables: “Enriched pathways”

Point 5. Page 12, line 313: “Table 5” may be corrected to “Table S5”.

Response 5. Thank you for the advice, we corrected with “Table S5” (Now Page 12, line 323)

Point 6. Pages 13-14: Sentences in lines 384 to 394 and those in lines 396 to 406 are completely same. It may be an error during editing the manuscript, and should be corrected.

Response 6. Thank you for the advice, we deleted the duplicated text (Now, Page 14,  lines 428-440)

Reviewer 2 Report

Comments and Suggestions for Authors

Authors have transplanted mouse melanoma tumor cells to mice and analyzed tumor samples grown for 9 days versus grown for 23 days. In each group 4 samples were used for nuclei isolation and subsequently single-nuclei transcriptome profiling has been performed.

Recently several single cell analysis manuscripts have been published (see EBioMedicine. 2024 Feb:100:104969. doi: 10.1016/j.ebiom.2024.104969.), but single-nuclei profiling is a novel and promising technology. Here are some points to consider:

11)      Only two groups are used for analysis. Early and late time points of melanoma growth in mice. How do we know that the difference in those two conditions reflects a useful readout?

It would be important to show that the cell-population changes, as well as gene-regulation changes, have predictive value for human melanoma progression. A human cohort should be used to test if the identified alterations have diagnostic value. TNM staging, Clark level, metastasis, therapy response are some characteristics which should be checked.

22)      The analysis could be extended by including Ligand–Receptor Cell–Cell Interactions. With software like e.g. cell-phone interaction between different cell types via ligands and their cognate receptors can be shown.

Minor points

1)    In the abstract days of harvest are described as 9 days and 24 days. In Figure 1 its 9 days and 9+14= 23 days.

2)    In Figure 2: is Etv1 one of the best melanoma markers?

3)    In Figure 3: 20 clusters are shown. The text refers to 21 clusters (line 161). Why the difference?

Author Response

Response to Reviewer #2

  1. Summary

Authors have transplanted mouse melanoma tumor cells to mice and analyzed tumor samples grown for 9 days versus grown for 23 days. In each group 4 samples were used for nuclei isolation and subsequently single-nuclei transcriptome profiling has been performed.

Recently several single cell analysis manuscripts have been published (see EBioMedicine. 2024 Feb:100:104969. doi: 10.1016/j.ebiom.2024.104969.), but single-nuclei profiling is a novel and promising technology.

Thank you for your positive assessment of our manuscript entitled “Single-nuclei transcriptome profiling reveals intra-tumoral heterogeneity and characterizes tumor microenvironment architecture in a murine melanoma model” (ID: ijms-3246868).  All the comments are valuable and very helpful for revising and improving our paper, as well as the important guiding significance to our research. We have studied comments carefully and have made correction which we hope meet with approval. Revised portion are marked in yellow in the paper. Main corrections and the answers to your comments are shown below.

  1. Questions for general evaluation

Does the introduction provide sufficient background and include all relevant references?

Yes/Can be improved/Must be improved/Not applicable

Are all the cited references relevant to the research?

Yes/Can be improved/Must be improved/Not applicable

Is the research design appropriate?

Yes/Can be improved/Must be improved/Not applicable

Are the methods adequately described?

Yes/Can be improved/Must be improved/Not applicable

Are the results clearly presented?

Yes/Can be improved/Must be improved/Not applicable

Are the conclusions supported by the results?

Yes/Can be improved/Must be improved/Not applicable

  1. Point-by-point reponses to Comments and Suggestions for Authors

Point 1. Only two groups are used for analysis. Early and late time points of melanoma growth in mice. How do we know that the difference in those two conditions reflects a useful readout? It would be important to show that the cell-population changes, as well as gene-regulation changes, have predictive value for human melanoma progression. A human cohort should be used to test if the identified alterations have diagnostic value. TNM staging, Clark level, metastasis, therapy response are some characteristics which should be checked.

Response 1. Thank you for reaching out about an predictive value for the human melanoma progression. However, we think that the suggested evalutaion is out the scope of the current manuscript, which is mainly to offer a proof of concept on the ability of sn-RNAseq to demonstrate the shape changes on TME cell populations in two different stages of a syngeneic melanoma mouse model.

In order to addres the request we added a paragraph at the end of the Discussion to underline the limit of our study and the need to validate in human melanomas the TME modification according to different progression stages:

Limitations of our study

Despite our promising observations suggesting that sn-RNAseq technique might  improve our understanding of melanoma biology, more detailed insights are required  to better profile the progression of tumor. Our snRNA-seq analysis done on 4 samples has simply demonstrated a clear change  in TME cell populations in  early stage and advanced stage of syngeneic D4M in vivo growth.  It is plausible that more samples could detect subtler differentiation within TME including the detection of minute cell populations.  Furthermore, any clinical translation of the observations here reported will require a validation in different progression stages of   human melanomas.    

Point 2. The analysis could be extended by including Ligand–Receptor Cell–Cell Interactions. With software like e.g. cell-phone interaction between different cell types via ligands and their cognate receptors can be shown.

Response 2. Thank you for this accurate suggestion. We improved our analysis and we included in the manuscripts results obtained with a “Ligand-receptor interaction” (new Figure 6; Page 12 , lines 340-350) . We used CellChat tools.

Minor points

Point 3. In the abstract days of harvest are described  as 9 day sand 24 days.  In Figure 1 it’s 9 days and 9+14= 23 days.

Response 3. We corrected the sentence in the abstract (Page 1, line 24) and the sentence in the Result section (Page 2, line 86)

Point 4. In Figure2: is Etv1 one of the best melanoma markers?

Response 4. Thanks to the reviewer's advice. It has been recently reported that a number of transcription factor of the ETS family were analyzed in multiple public datasets, comparing their expression in melanoma versus the normal skin. The authors reported that ETV, together with ELK3, ETS1, ETV4, ETV5, is significantly upregulated in melanoma (Qu H et al. 2020, Frontiers in Immunology; doi: 10.3389/fimmu.2020.612784). We added the reference of this paper in the Results section when we indicated the use of this marker for melanoma cells/nuclei (Page 4, line 132-133)

Point 5. In Figure 3: 20 clusters are shown. The text refers to 21 clusters (line161). Why the difference?

Response 5. Thank you very much for your advice. However, in the figure 3 we show 21 clusters, clusters are named from c0 to c20.

Reviewer 3 Report

Comments and Suggestions for Authors

Authors provide here single-nuc. RNAseq data of a murine melanoma (D4M) model and demonstrated here the heterogeneity of tumors. Gaining insights into the tumor heterogeneity is of high importance as it provides information on the cellular subtypes and potentially identifies druggable pathways. Although the study´s results are well-presented, I have the feeling that the claim of changes in the composition of tumor and immune cells needs to be proven or somehow underpinned by additional data or discusses as a weakness of the study.    

1.)    Knowing the extreme heterogeneity from spatial transcriptomics profiling not only among but also within tumors, authors must indicate the number of biologic replicates which have been investigated. If authors investigated two replicates of either state (early and late) authors should discuss that the observed heterogeneity likely not only reflects the different states but also the heterogeneity among tumors.

2.)    On page 2, line 92: the term integrity might be a more accurate term than viability

3.)    Authors should add a declaration that mouse studies have been proven by a local ethics committee.

4.)    In line 113 (page 4) authors should provide information on the number of tumors per investigation (early, late).

5.)    In lines 124, 125 authors claim an …”overall increase in malignant cells and in the myeloid cell in the advanced stage 125 while fibroblasts, T cells endothelial and lymphatic endothelial cell populations decreased” Did authors take into consideration that this may simple reflect the normal tumor heterogeneity or simply reflects the increased tumor size/distribution of immune cells within the tumor? Maybe a immunohistochemically staining for immune cells such as CD3+ T cells may strengthen this finding. If tumors simply grow and are not influenced by external stress situations such as changes in the immune cell activation or therapy-induced stress I would not expect drastic changes in the tumor architecture, particularly syngeneic tumors should not only show changes in he cellular composition reflected by tumor growth. As tumors grow for 24 days without drug-induced selection processes I would also not expect a drastic diverging evolution of genetic subclones. So, also should discuss these aspects.

6.)    The information on the sample size should be provided earlier. UMAP plots shown in Figure 3b somehow indicate that the cellular composition of tumors does not change drastically but remain quite stable. The change in tumor cells/immune cell numbers might reflect the different tumor size (more tumor cells in the advanced stage tumors) but less immune cells (which might reflect the immune suppressive mechanisms preventing a homogeneous distribution of immune cells with the tumors.

7.)    In Figure 4B authors should include information on the types of clusters (which are explained in b not a), do all clusters reflect different tumor cell types or tumor cells, as I did not find any classical markers for endothelial cells such as Vwf, Pecam1, Cdh5, Epas1 etc. they are not included here?

8.) Authors should also discuss that the low sample size (I am aware that snRNAseq is still an expensive method) may also not support the claim of tumor/immune cell changes by tumor growth.

In summary, authors provided here a well-presented proof-of-principle of single-nuclei RNAseq of murine tumors and dissected their cell types. However, authors should be very careful with statements regarding the compositional changes of tumors and should include statistics which have not been included yet.

Comments on the Quality of English Language

A generall check of the structure of sentences should be performed.

Author Response

Response to Reviewer #3

  1. Summary

Authors provide here single-nuc. RNAseq data of amurine melanoma (D4M) model and demonstrated here the heterogeneity of tumors. Gaining insights into the tumor heterogeneity is of high importance as it provides information on the cellular subtypes and potentially identifies druggable pathways. Although the study ́s results are well-presented, I have the feeling that the claim of changes in the composition of tumor and immune cells needs to be proven or somehow underpinned by additional data or discusses as a weakness of the study.

Thank you for your positive comments concerning our manuscript entitled “Single-nuclei transcriptome profiling reveals intra-tumoral heterogeneity and characterizes tumor microenvironment architecture in a murine melanoma model” (ID: ijms-3246868). All the comments have been valuable and very helpful for revising and improving the manuscript, as well as the important guiding significance to our research. We have carefully analyzed your comments and we made corrections and/or modifications, which we hope meet with approval. Revised portions are marked in yellow in the paper. The main corrections/modifications made and the answers to your comments are shown below.

  1. Questions for general evaluation

Does the introduction provide sufficient background and include all relevant references?

Yes/Can be improved/Must be improved/Not applicable

Are all the cited references relevant to the research?

Yes/Can be improved/Must be improved/Not applicable

Is the research design appropriate?

Yes/Can be improved/Must be improved/Not applicable

Are the methods adequately described?

Yes/Can be improved/Must be improved/Not applicable

Are the results clearly presented?

Yes/Can be improved/Must be improved/Not applicable

Are the conclusions supported by the results?

Yes/Can be improved/Must be improved/Not applicable

Point-by-point reponses to Comments and Suggestions for Authors

Point1. Knowing the extreme heterogeneity from spatial transcriptomics profiling not only among but also within tumors, authors must indicate the number of biologic replicates which have been investigated. If authors investigated two replicates of either state (early and late) authors should discuss that the observed heterogeneity likely not only reflects the different states but also the heterogeneity among tumors.

Response 1: Thank you very much for your advice. We investigated 2 biological replicates for each condition, early and advanced stage for a total of 4 tumor analyzed. We added the total number of tumors analyzed in Results section (Page 2, line 88).

In order to addres the request we added a paragraph at the end of the Discussion section:

Limitations of our study

Despite our promising observations suggesting that sn-RNAseq technique might  improve our understanding of melanoma biology, more detailed insights are required  to better profile the progression of tumor. Our snRNA-seq analysis done on 4 samples has simply demonstrated a clear change  in TME cell populations in  early stage and advanced stage of syngeneic D4M in vivo growth.  It is plausible that more samples could detect subtler differentiation within TME including the detection of minute cell populations.  Furthermore, any clinical translation of the observations here reported will require a validation in different progression stages of  human melanomas.    

Point 2. On page 2,line 92: the term integrity might be a more  accurate term than viability.

Response 2. Thank you very much for the kind suggestion. We replaced the term viability with the word “integrity”, as suggested (Page 2, line 92).

Point 3. Authors should add a declaration that mouse studies have been proven by a local ethics committee.

Response 3. Thank you very much for your reminding. We added the sentence “All animal procedures were approved by Italian Ministry of Health (protocol 21635.13) and were performed in accordance with institutional guidelines and international law and policies” in the matherial and methods section (4.1 Sample description; Page 14-15, lines 460-462).

Point 4.  In line 113 (page 4) authors should provide information on the number of tumors per investigation (early, late).

Response 4. Thank you for the advice. We modify the sentence in: “Sn-RNAseq data from 4 samples (2 specimens from early stage condition and 2 specimens from advanced stage condition) was analyzed by Seurat” (Page 4, lines 113-114).

Point 5.  In lines 124, 125 authors claim an ...”overall increase in malignant cells and in the myeloid cell in the advanced stage 125 while fibroblasts, T cells endothelial and lymphatic endothelial cell populations decreased” Did authors take into consideration that this may simple reflect the normal tumor heterogeneity or simply reflects the increased tumor size/distribution of immune cells within the tumor? Maybe a immunohistochemically staining for immune cells such as CD3+ T cells may strengthen this finding. If tumors simply grow and are not influenced by external stress situations such as changes in the immune cell activation or therapy-induced stress I would not expect drastic changes in the tumor architecture, particularly syngeneic tumors should not only show changes in he cellular composition reflected by tumor growth. As tumors grow for 24 days without drug-induced selection processes I would also not expect a drastic diverging evolution of genetic subclones. So, also should discuss these aspects.

Response 5.

Thank you for this accurate suggestion. In order to confirm the reduction in tumor infiltration of T cells that has been revealed by our analysis in sn-RNAseq dataset, we analyzed by flow cytometry the tumor infiltration of CD4+ and CD8+ population in D4M melanoma tumors in early stage and in advanced stages. The figures below show these data and suggest that in this specific melanoma tumor model, T cell number decreases significantly during the tumor progression, indicating a mechanism of tumor suppression that is associated to the in vivo tumor growth. These data have been inserted as a new supplementary figure, Figure S2. In the revised version of the manuscript, we added the sentence “Flow cytometry data on confirmed the significant reduction of   CD4+ and CD8+ T cell population in advanced stage tumors compared to early stage D4M melanoma tumors (Figure S2).” (Page 4; lines 128-130). Furthermore, we added in the Materials and Methods section a paragraph “4.9 Flow Cytometry”, that described the method used for the flow cytometry analysis.

Supplementary Figure 2. T lymphocytes tumor infiltration in D4M melanoma tumors in early stage and in advanced stages. (A) Representative flow cytometry plot and quantification of tumor infiltrating CD45+CD4+ T lymphocytes in early stages and advanced stages D4M melanoma tumors.  (B) Representative flow cytometry plot and quantification of tumor infiltrating CD45+CD8+ T lymphocytes in early stages and advanced stages D4M melanoma tumors.

Point 6.  The information on the sample size should be provided earlier. UMAP plots shown in Figure 3b somehow indicate that the cellular composition of tumors does not change drastically but remain quite stable. The change in tumor cells/immune cell numbers might reflect the different tumor size (more tumor cells in the advanced stage tumors) but less immune cells (which might reflect the immune suppressive mechanisms preventing a homogeneous distribution of immune cells with the tumors)

Response 6. Thanks to the reviewer’s suggestion. UMAP plot in figure 3B visualized in each dataset cluster the cell abundance in early stage (upper panel) and in advanced stage condition (bottom panel). However, the quantification for each cluster is shown in Table S2 that indicates the specific cell redistribution during tumor growth/progression. Actually, some specific clusters were dramatically changed between the 2 conditions. E.g. cluster c0: 247 nuclei (3.2%) in early stages and 1604 (17.7%) in advanced stage; cluster c12: 402 nuclei (5.2%) in early stages and 171 (1.9%) in advanced stage.

Point 7. In Figure 4B authors should include information on the types of clusters (which are explained in b not a), do all clusters reflect different tumor cell types or tumor cells, as I did not find any classical markers for endothelial cells such as Vwf, Pecam1, Cdh5, Epas1 etc. they are not included here?

Response 7. Thanks to the reviewer’s comments. In Figure 4 we focused our attention only in clusters associated to melanoma cancer cells to discuss the effect of the tumor growth/progression on intratrumoral heterogeneity. In figure 4B we showed selected enriched genes associated only to melanoma cancer cells clusters. We changed the sentence in Figure 7 legend: “… selected enriched genes used for biological identification of melanoma cancer cells heterogeneity among each cluster associated to melanoma cancer cells”  (Page 9, line 305). The full list of the top 20 most upregulated genes in each clusters (c0-c20) reported in Table S3. In this table Cluster 17, that is associated to endothelial cells,  show the enrichment in genes that are marker of endothelial cells like Mecom, Col4a1, Flt1, Pecam1.

Point 8. Authors should also discuss that the low sample size (I am aware that snRNAseq is still an expensive method) may also not support the claim of tumor/immune cell changes by tumor growth.

Response 8.

In order to addres the request we added a paragraph at the end of the Discussion section:

Limitations of our study

Despite our promising observations suggesting that sn-RNAseq technique might  improve our understanding of melanoma biology, more detailed insights are required  to better profile the progression of tumor. Our snRNA-seq analysis done on 4 samples has simply demonstrated a clear change  in TME cell populations in  early stage and advanced stage of syngeneic D4M in vivo growth.  It is plausible that more samples could detect subtler differentiation within TME including the detection of minute cell populations.  Furthermore, any clinical translation of the observations here reported will require a validation in different progression stages of  human melanomas.    

Point 9. In summary, authors provided here a well-presented proof-of-principle of single-nuclei RNAseq of murine tumors and dissected their cell types. However, authors should be very careful with statements regarding the compositional changes of tumors and should include statistics which have not been included yet.

Response 9. Thank you very much for your advice. We added in Material and Methods section the paragraph:

4.10 Statistical analysis

Statistical analyses of snRNA-seq data were performed by CellBender, Seurat (v5.1.0) and CellChat. Most of the analyses were done in Python (version 3.11.7) and R (version 4.4.0) software.And p-value below 0.05 was considered statistically significant in this research. Flow cytometry analysis was performed by Student’s t-test.

Round 2

Reviewer 2 Report

Comments and Suggestions for Authors

The authors have addressed both major points.

Allthough point 1 could not be fullfilled, at least reasons are clearly stated in the manuscript.

Cellchat generated figures enrich the manuscript. A myeloid and T-cell interaction is visible.

Reviewer 3 Report

Comments and Suggestions for Authors

Thanks for adressing my concerns.

Comments on the Quality of English Language

quality of language is fine, however authors should read and seek for mistakes that have been made e.g. structure of sentences